# High Expression of Tetraspanin 5 as a Prognostic Marker of Colorectal Cancer

**DOI:** 10.3390/ijms24076476

**Published:** 2023-03-30

**Authors:** Sanghyun Roh, Sooyoun Kim, Inpyo Hong, Minho Lee, Han Jo Kim, Tae Sung Ahn, Dong Hyun Kang, Moo-Jun Baek, Hyoung Jong Kwak, Chang-Jin Kim, Dongjun Jeong

**Affiliations:** 1Department of Pathology, College of Medicine, Soonchunhyang University, 31 Soonchunhyang 6 gil, Dongnam-gu, Cheonan 31151, Chungcheongnam-do, Republic of Koreasooy.kim@sch.ac.kr (S.K.);; 2Department of Oncology, College of Medicine, Soonchunhyang University, 31 Soonchunhyang 6 gil, Dongnam-gu, Cheonan 31151, Chungcheongnam-do, Republic of Korea; 3Department of Surgery, College of Medicine, Soonchunhyang University, 31 Soonchunhyang 6 gil, Dongnam-gu, Cheonan 31151, Chungcheongnam-do, Republic of Korea; 4Research Institute of Clinical Medicine, Woori Madi Medical Center, 111 Baekjedae-ro, Wansan-gu, Jeonju 55082, Jeollabuk-do, Republic of Korea

**Keywords:** colorectal cancer, prognostic biomarker, therapeutic target, biomarker, TSPAN5

## Abstract

Cancer is a major disease and the leading cause of death worldwide, with colorectal cancer (CRC) being the third-most common cancer in Korea. The survival rate associated with CRC reduces as the disease stage increases. Therefore, its early detection and treatment can greatly increase patient survival rates. In this study, we identified the tetraspanin 5 (*TSPAN5*) gene as an important biomarker for predicting the prognosis of patients with CRC. A TMA slide was used for statistical analysis. pN and clinical stage were found to be significant factors according to chi-square analysis, whereas pT, pN, metastasis, clinical stage, and TSPAN5 expression were significant according to Cox regression analysis. In order to prove the usefulness of TSPAN5, which is overexpressed in patients with metastatic CRC, as a biomarker, proliferation, migration, invasion, and tumorigenicity were examined using cell lines inhibited using small interfering RNA. The evaluations confirmed that *TSPAN5* suppression, in turn, suppressed proliferation, migration, invasion, and tumorigenesis, which are characteristic of cancer cells. Therefore, the evaluation of TSPAN5 expression may help observe the prognosis of CRC and determine an appropriate treatment method for patients with CRC.

## 1. Introduction

Colorectal cancer is the third-most common cancer worldwide [1,2]. According to the Korea National Statistical Office, the mortality rate associated with colorectal cancer is 93.9% for localized metastasis, 82.1% for regional metastasis, and 19.8% for distant metastasis [3,4]. Surgical resection is the optimal treatment option for colorectal cancer, but tumor recurrence after resection is associated with a high risk of cancer-related death. At the first diagnosis, approximately two-thirds of patients with colorectal cancer undergo resection for therapeutic purposes; however, 30–50% of these patients experience cancer recurrence and eventually die [5,6]. Therefore, the development of prognostic biomarkers that can predict recurrence and metastasis in patients undergoing surgery for colorectal cancer is extremely important. Currently, KRAS and BRAF are used as prognostic biomarkers for colon cancer, but they are expressed in only 7–45% of all patients with colorectal cancer [7]. The prediction of the recurrence of colorectal cancer is a critical step in determining future treatments, such as adjuvant chemotherapy and targeted molecular therapy, which can increase the sensitivity of the existing treatments. However, the mortality rate of patients with colorectal cancer remains relatively high owing to the current absence of reliable prognostic biomarkers. In a previous study, we aimed to identify biomarkers related to the recurrence and metastasis of colorectal cancer, and tetraspanin 5 (*TSPAN5*) was selected as a candidate for the present study.

Tetraspanins (*TSPANs*) are a family of membrane proteins characterized by four membrane-penetrating spirals. Tetraspanins are abundant in the cell membrane and are also present in the organelles and granules of almost all cells and tissues [8,9]. Unlike most other cell surface proteins, TSPANs have not yet clearly been identified as receptor molecules. Several reports have reported the direct interaction of TSPANs with cells or soluble ligands. For example, pregnancy-specific glycoprotein 17, a protein secreted in the placenta of mice, binds to TSPAN29 (CD9) in macrophages to secrete anti-inflammatory cytokines [10]. In one study, the Duffy antigen receptor for chemokines (DRAK), which is thought to be essential for the functioning of CD82 as a metastatic inhibitor, was identified as an interactive partner of TSPAN27 (CD82) [11]. Through the components of tetraspanin-enriched microdomains, TSPANs exert strong effects on cell adhesion, migration, invasion, and signaling; cell-cell fusion; and viral infection, all of which play important roles in cancer development [9,12,13,14]. For example, some TSPANs, such as TSPAN24 (CD151), interact with different integrins to regulate cell adhesion to different layers of epithelial or endothelial cells, which in turn affects cell morphology, migration, and signaling [15,16]. 

TSPAN5 mediates signaling events that play a role in the regulation of cell development, activation, growth, and motility in normal tissues, and it has been shown to interact with ADAM10 [17]. TSPAN5 has a regulatory effect on the transport of ADAM10 from vesicles to the protoplasmic reticulum, and regulatory disturbances in these interactions may be associated with disorders that cause cancer, Alzheimer’s disease, or inflammation [18,19]. In cancerous tumors, TSPAN5 plays a role in hepatocellular carcinoma (HCC) metastasis through epithelial-mesenchymal transition [20] and regulates the growth of HCC in co-operation with other genes [21]. However, its role in colorectal cancer has not been clearly identified.

In this study, we investigated the role of TSPAN5 in colorectal cancer recurrence and metastasis using colorectal cancer cell lines and tissues from patients with colorectal cancer.

## 2. Results

### 2.1. Confirmation of TSPAN5 Expression in Colon Cancer Cell Lines

TSPAN5 mRNA expression was quantified based on the expression value of glyceraldehyde-3-phosphate dehydrogenase (GAPDH), and the relative expression levels in different cell lines were compared. Compared with SW620 cells, which showed the highest expression of TSPAN5, the SW480 cells showed 14.5% lower expression, HCT116 cells showed 17.3% lower expression, and the HT29 cells showed 22.3% lower expression (Figure 1A,B).

TSPAN5 protein expression levels were quantified based on the expression value of β-actin, and the relative expression levels in different cell lines were compared. Compared with SW620 cells, which showed the highest expression of TSPAN5, SW480 cells showed 8.3% lower expression, HCT116 cells showed 9.2% lower expression, and the HT29 cells showed 28.0% lower expression (Figure 1C,D). The findings confirmed that the HT29 cells showed lower TSPAN5 mRNA and protein expression than the other cancer cell lines. Therefore, subsequent experiments were conducted using the SW480 and SW620 cells, which showed high mRNA and protein expression levels.

### 2.2. Confirmation of Gene Silencing by siTSPAN5

Reverse-transcription polymerase chain reaction (RT-PCR) was used to confirm the expression level of TSPAN5 relative to that of GAPDH. After transfection with siTSPAN5, the SW480 cells showed a 34.0% reduction compared with those in the control group, whereas the SW620 cells showed a 41.2% reduction (Figure 2A,B). The results obtained using western blotting confirmed the suppression of *TSPAN5*, showing a decrease of 44.7% in the SW480 cells and 52.5% in the SW620 cells compared with that in the control cells (Figure 2C,D).

### 2.3. Reduction of Cancer Cell Proliferation by Inhibiting TSPAN5 Expression

The average absorbance measured for 72 h in the SW480 cells was 0.315 for the siTSPAN5 group, representing a 67.6% reduction compared with that in the control group (0.963) (Figure 3A,B). In the SW620 cells, the average absorbance in the siTSPAN5 group was 0.370, a 63.6% reduction compared with that in the control group (1.018) (Figure 3C,D). 

### 2.4. Reduction of Cancer Cell Migration and Invasion Ability by Inhibiting TSPAN5 Expression

In our analyses of the average number of cells and the rate of decrease, the migration ability of the SW480 cells decreased by 60.8%, from 1271.0 cells in the control group to 497.6 cells in the siTSPAN5 group. In the SW620 cells, the migration ability decreased by 84.3%, from 1220.6 cells in the control group to 192.2 cells in the siTSPAN5 group (Figure 4A,B). The invasion ability of the SW480 cells decreased by 67.4%, from 254.0 in the control group to 82.8 in the siTSPAN5 group. In the SW620 cells, the invasion ability decreased by 70.0%, from 210.6 in the control group to 63.2 in the siTSPAN5 group (Figure 4C,D).

### 2.5. Reduction of Colony Formation Ability by Inhibition of TSPAN5 Expression

The average number and reduction rate of colonies indicated a reduction of 52.7% in the SW480 cells, from 142.8 in the control group to 67.6 in the siTSPAN5 group, and a reduction of 65.8% in the SW620 cells, from 212.5 in the control group to 72.6 in the siTSPAN5 group (Figure 5A,B).

### 2.6. Confirmation of TSPAN5 Gene Expression through Immunohistochemical Staining in Colorectal Cancer Tissue

*TSPAN5* was not well-expressed in the cells other than the background in the normal tissues but was strongly expressed at the location of the cell membrane in the colon cancer tissues (Figure 6).

### 2.7. Relationship between TSPAN5 Expression Level and Clinicopathological Factors of Colorectal Cancer

Among the 200 tissues of the patients with colorectal cancer, TSPAN5 was highly expressed in 148 (74%), and 52 tissues showed low expression (26%). For statistical verification, the correlations of TSPAN5 expression with age, sex, pT stage, pN stage, metastasis, and the clinical stage of the patient were examined. TSPAN5 expression was not significantly correlated with metastasis but was found to be significantly correlated with sex (*p* = 0.034), pN stage (*p* < 0.001), and clinical stage (*p* = 0.005) (Table 1).

### 2.8. Univariate and Multivariate Cox Regression Models and Patient Outcomes

In the univariate Cox regression analysis, pT stage (HR [hazard ratio] = 3.051, 95% confidence interval [CI] = 1.249–7.455, *p* = 0.014), pN stage (HR = 2.051, 95% CI = 1.459–2.885, *p* < 0.001), metastasis (HR = 1.623, 95% CI = 1.159–2.271, *p* = 0.005), clinical stage (HR = 2.697, 95% CI = 1.864–3.903, *p* < 0.001), and TSPAN5 expression (HR = 1.763, 95% CI = 1.220–2.549, *p* = 0.003) were evaluated as significant, and in the multivariate Cox regression analysis, metastasis (HR = 1.911, 95% CI = 1.608–2.363, *p* = 0.049), clinical stage (HR = 4.078, 95% CI = 1.969–8.445, *p* < 0.001), and TSPAN5 expression (HR = 1.528, 95% CI = 1.027–2.274, *p* = 0.036) were shown to be significant (Table 2).

### 2.9. Relationship between TSPAN5 Expression and Patient Survival

The survival rate of the low-expression group (*n* = 52) gradually declined and was approximately 54% after 5 years. In contrast, the survival rate of the high-expression group rapidly decreased to approximately 38% after 5 years, which was significantly lower than that of the low-expression group (Figure 7).

## 3. Discussion

According to the World Cancer Report published by the International Institute of Cancer (IARC) in 2020, cancer is the leading cause of death worldwide, with 18 million new cancer cases diagnosed in 2020, and colorectal cancer is the third-most diagnosed cancer worldwide [1,2]. The TNM staging system is mainly used to determine colorectal cancer prognosis, but there are no reliable prognostic biomarkers that can reflect the cellular heterogeneity in tumors. Therefore, prognostic biomarkers should be identified to improve the prognosis and predict recurrence and metastasis. *TSPAN5* has been shown to interact with ADAM10 [17] and act as an oncogene in HCC [20,21] and a tumor suppressor gene in gastric cancer [22]. Thus, we hypothesized that *TSPAN5* is a key gene in mediating signaling events that play a role in the regulation of cell development, activation, growth, and motility in normal tissues.

In this study, the location of *TSPAN5* expression was confirmed through immunohistochemistry staining using cancer tissue, and the results of various statistical analyses, such as chi-square, Cox regression, and Kaplan–Meier analyses, indicated that TSPAN5 expression increased with the progression of colorectal cancer. The significance of the relationships regarding TSPAN5 expression was proven for pT, pN, clinical stage progression, and metastasis. In this study, the overexpression of TSPAN5 was confirmed to be an independent prognostic marker that was associated with reduced overall survival in patients with colorectal cancer.

We investigated the role of TSPAN5 in the functional aspects of cancer by suppressing the gene using small interfering RNA (siRNA), focusing on the colorectal cancer cell lines SW480 and SW620. The siTSPAN5 group showed lower proliferation, migration, invasion, and tumorigenicity of cancer cells than the control group. These findings highlighted the possibility of reducing the cancer functions related to recurrence and metastasis through TSPAN5 gene suppression and using TSPAN5 as a therapeutic target in colorectal cancer.

Some researchers mentioned that TSPAN5 is associated with ADAM10, which plays a vital role in the Notch signaling pathway, and the Rubinstein group showed that TSPAN5 is important for Notch signaling in cell line models [19,23,24,25,26]. Therefore, future studies should aim to assess the effects of TSPAN5 and other signal transduction systems in colorectal cancer and clarify the relationship between tumor formation and metastasis on the basis of the expression of TSPAN5 through animal experiments. Although studies on TSPAN are steadily progressing, few studies have evaluated TSPAN and tumors. Furthermore, a previous study has described TSPAN5 overexpression in colorectal cancer and suggests the potential of TSPAN5 as a biomarker for colorectal cancer [27]. Therefore, this study is significant as it is, to the best of our knowledge, the first to show the potential of TSPAN5 as a biomarker in colorectal cancer and to reveal that high TSPAN5 expression is an independent prognostic factor for colorectal cancer and can be helpful as a future treatment target.

## 4. Materials and Methods

### 4.1. Cell Lines and Cell Culture

The human CRC cell lines SW480, SW620, HCT116, and HT29 were obtained from the Korean Cell Line Bank (Seoul, Republic of Korea). The four cell lines were grown at 37 °C in Roswell Park Memorial Institute 1640 medium (Welgene, Gyeongsan, Republic of Korea) supplemented with 10% heat-inactivated fetal bovine serum (Thermo Fisher, Waltham, MA, USA) and 1% penicillin-streptomycin (Corning, NY, USA) in a humidified incubator containing 5% CO_2_ at 37 °C.

### 4.2. siRNA Transfection Assay

TSPAN5 siRNA was produced by Bioneer (Daejeon, Republic of Korea) to inhibit TSPAN5 expression. Two types of colorectal cancer cell lines, SW480 and SW620, were seeded on a 6-well plate and incubated for 24 h. Subsequently, cell medium was removed, and serum-free medium was added. Lipofectamine RNAiMAX (Waltham, MA, USA) was added to opti-MEM (Thermo Fisher, Waltham, MA, USA), a mixture reacted at room temperature for 5 min, and TSPAN5 siRNA was added to opti-MEM, and the mixture was mixed and incubated for 20 min. Next, the mixture was added to each well, the plate was shaken to mix evenly, and the mixture was cultured in a 5% CO_2_ incubator at 37 °C to induce gene expression inhibition. After 24 h, it was replaced with a growth medium containing fetal bovine serum (FBS), and cell stabilization was performed for 24 h. Equal amounts of transfection reagent were used to evaluate the cytotoxicity of the reagent. The siRNA sequences were as follows: TSPAN5(1) 5′-GAG CAU AUC GGG AUG ACA U[dT][dT]-3′ and TSPAN5(2) 5′-AUG UCA UCC CGA UAU GCU C[dT][dT]-3′.

### 4.3. Reverse-Transcription Polymerase Chain Reaction

RNA was extracted using RiboEx^TM^ (GeneAll Biotechnology, Seoul, Republic of Korea), and cDNA was synthesized using HyperScript^TM^ RT Master Mix (GeneAll Biotechnology, Seoul, Republic of Korea) with 500 ng of RNA according to the manufacturer’s instructions. The reaction was performed at 50 °C for 60 min and then treated at 85 °C for 5 min to stop the reaction. The PCR process for mRNA expression was repeated 38 times in the following order: pre-denaturation (95 °C for 10 min), denaturation (95 °C for 15 s), annealing (60 °C for 30 s), and extension (72 °C for 30 s). The relative expression level of each mRNA was quantified using GAPDH. The sequences of the primers used in PCR were as follows: TSPAN5 forward, 5′-GAGCGCTACGGGAAAACACT-3′; TSPAN5 reverse, 5′-CCACAGCACTGCCAATATTCCT-3′ (220 bp); GAPDH forward, 5′-CACTACCAAGGACAAGGCGTTC-3′; and GAPDH reverse, 5′-CAACGCCTCTTTGGTCTCCTTG-3′ (151 bp). PCR was performed on a 2% agarose gel containing NEOgreen DNA staining reagents (Neoscience, Seoul, Republic of Korea). The results were confirmed using the FluoroBox nucleic acid gel imaging system (Neoscience, Seoul, Republic of Korea), and the relative mRNA expression of GAPDH was analyzed using ImageJ.

### 4.4. Western Blot Assay

Total protein was extracted using the PRO-PREP^TM^ Protein Extraction Solution (C/T) (iNtRON, Seongnam, Republic of Korea). Protein was quantified (30 μg/lane) with the Pierce^TM^ BCA Protein Assay Kit (Thermo Fisher Scientific, CA, USA) using Multiskan^TM^ GO (Thermo Fisher Scientific, CA, USA). Ten percent of sodium dodecyl sulfate polyacrylamide gel was used for electrophoresis, which was followed by transfer to a 0.2-μm polyvinylidene difluoride membrane (Merck Millipore, MA, USA). The primary antibody (anti human TSPAN5 mouse monoclonal antibody [TS-3], 1:5000, gifted by Dr. Rubenstein, France) and anti β-actin antibody (C4) (#sc-47778, 1:1000, Santa Cruz Biotechnology, CA, USA) were treated with a secondary antibody (anti mouse IgG [H+L], HRP conjugated [#W4021], 1:5000, Promega, WI, USA). After confirmation using the ChemiDoc XRS+ System (Bio-Rad Laboratories, CA, USA), the protein expression with reference to β-actin was analyzed using ImageJ. 

### 4.5. Proliferation Assay

Cells (density: 1 × 10^4^) of the control and siTSPAN5 groups were divided into 96 well plates in triplicate and incubated at 37 °C in a 5% CO_2_ incubator for 24, 48, and 72 h. Their absorbance was then measured at 450 nm using MultiskanTM GO (Thermo Fisher Scientific, Carlsbad, CA, USA) to evaluate cell proliferation.

### 4.6. Migration and Invasion Assays

Migration and invasion assays were performed using a 24 well plate containing 6.5 mm Transwell^®^ with 8.0 µm Pore Polycarbonate Membrane Insert (Corning, NY, USA). Invasion assays were performed using Corning^®^ Matrigel^®^ Matrix (5 mg/mL; Corning, NY, USA) dispensed on a Transwell insert and then solidified in an incubator for 1 h before use. Control and siTSPAN5 cells at a density of 3 × 10^5^ were dispensed into the Transwell insert. The growth medium was added outside the insert and incubated. Migration and invasion were induced for 48 and 72 h, respectively, in an incubator. Thereafter, 4% paraformaldehyde was used to fix the cells. The cells were then permeabilized with 100% methanol. Cell staining was performed using 0.005% crystal violet, and the stained cells were counted after randomly selecting five areas under a microscope.

### 4.7. Anchorage-Independent Colony Formation Assay

Agarose (Bio-D, Wangmyeong, Republic of Korea) was added to Dulbecco’s phosphate buffered saline (DPBS) and dissolved with microwave to prepare 1% agarose, and the temperature was adjusted in a 40 °C water bath with a growth medium containing 20% FBS. The two solutions were mixed in a 1:1 ratio; finally, 0.5% agar was prepared and divided for 1.5 mL on a 6 well plate. The bottom agar was completely solidifying at 4 °C. Using the same method, 20% FBS medium and 0.7% agarose, which were temperature adjusted in a 40 °C water bath, were mixed with 0.35% top agar; next, 1 × 10^4^ cells of the experimental group, which were suppressed by siRNA and control group cells, were prepared per well and then mixed with top agar. Then, 1.5 mL of top agar mixture was divided on the bottom agar, separated at room temperature, and then cultured in a 5% CO_2_ incubator at 37 °C. The size of the colony of the control cells was checked every week, and 1 mL of the growth medium was placed on top of the top agar so that the agar did not dry. After a 2 week incubation period, 0.005% crystal violet was divided on the top agar, reacted at room temperature for 1 h, and washed with DPBS. Five zones were then randomly selected under the microscope, and the stained colony in the zone was counted.

### 4.8. Immunohistochemistry Assay

TMA slides containing tissues from 200 patients with colon cancer were obtained from Superbiochips (Seoul, Republic of Korea). TMA slides were retrieved and incubated with a primary antibody (anti human TSPAN5 mouse monoclonal antibody [TS-2], 1:50, gifted by Dr. Rubenstein, France) at 4 °C, followed by incubation with a secondary antibody (anti mouse IgG (H+L), HRP conjugated, #W4021, 1:100, Promega, WI, USA) at room temperature. To confirm the expression level of TSPAN5, diaminobenzidine (Dako, Denmark) was used, and counterstaining was performed using hematoxylin. Quantification of TSPAN5 was performed by assessing the percentage and intensity of stained tumor cells by two pathologists in a blinded manner. The staining percentage was scored as follows: 0, 0–5%; 1, 5–25%; 2, 25–50%; 3, 50–75%; and 4, 75–100%. Staining intensity was scored as follows: 0, negative; 1, weak; 2, moderate; and 3, strong. The final score was calculated by multiplying the scores for the two criteria (low, score < 4; high, score ≥ 4).

### 4.9. Statistical Analysis

SPSS (version 23.0; IBM, Armonk, NY, USA) was used for statistical analyses, and data are expressed as mean ± standard deviation. Relationships between clinicopathological factors and TSPAN5 expression were analyzed using the chi-squared test. Cox regression analysis was performed to calculate the risk between clinicopathological factors and genes, and 95% CI was estimated accordingly. The 5 year survival rate of patients with colorectal cancer was analyzed using Kaplan–Meier curves, and statistical significance was evaluated using the log rank test. Differences were considered statistically significant at *p* < 0.05. 

## Figures and Tables

**Figure 1 ijms-24-06476-f001:**
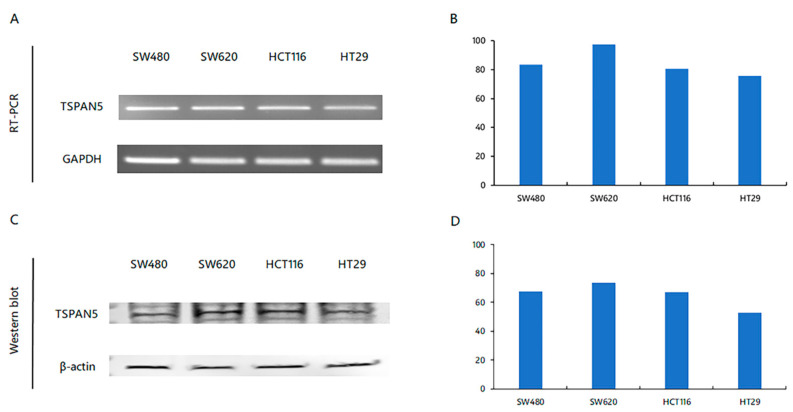
Expression of TSPAN5 in colorectal cancer cell lines. (**A**) Confirmation of TSPAN5 expression in cell lines determined using RT-PCR. (**B**) Graph of relative TSPAN5 mRNA expression level in cell lines determined using RT-PCR. (**C**) Confirmation of TSPAN5 expression of cell lines determined using western blotting. (**D**) Relative TSPAN5 protein expression in cell lines was determined using Western blotting.

**Figure 2 ijms-24-06476-f002:**
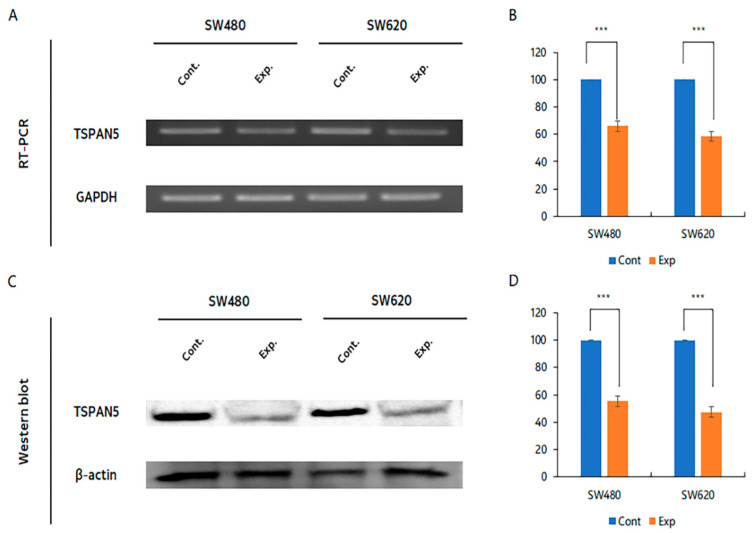
Confirmation of inhibition of colon cancer cell lines by siTSPAN5. (**A**) Confirmation of cell line inhibition using RT-PCR. (**B**) Relative mRNA expression graph between control and siTSPAN5 cell lines (*** *p* < 0.001). (**C**) Confirmation of cell line inhibition using Western blotting. (**D**) Relative protein expression graph between control and siTSPAN5 cell lines (*** *p* < 0.001).

**Figure 3 ijms-24-06476-f003:**
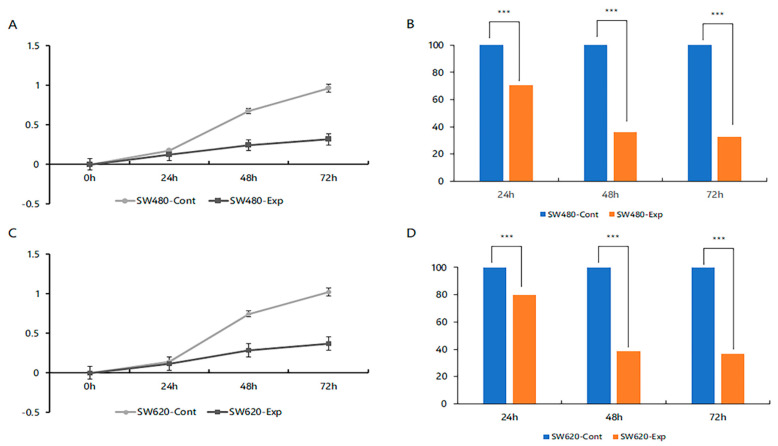
Proliferation test by siTSPAN5. (**A**,**B**) Proliferation graph of SW480 control and siTSPAN5 cells. (**C**,**D**) Proliferation graph of SW620 control and siTSPAN5 cells (*** *p* < 0.001).

**Figure 4 ijms-24-06476-f004:**
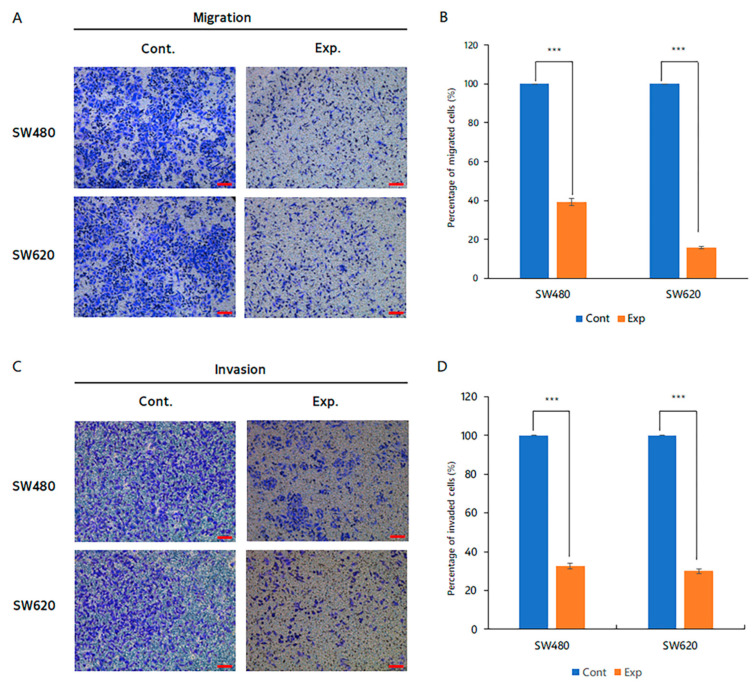
Migration and invasion confirmation test by siTSPAN5. (**A**) Evaluation of SW480 and SW620 cell migration ability. (Scale bar 100 μm). (**B**) Relative migration rate graph of each cell line (*** *p* < 0.001). (**C**). Evaluation of SW480 and SW620 cell invasion ability. (Scale bar 100 μm). (**D**) Relative invasion rate graph of each cell line rate (*** *p* < 0.001).

**Figure 5 ijms-24-06476-f005:**
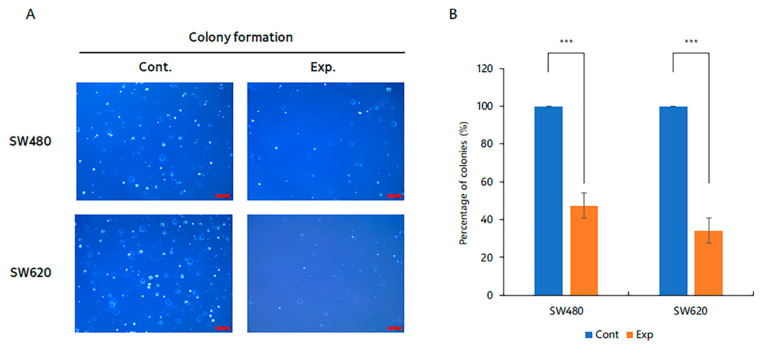
Tumorigenicity confirmation test by siTSPAN5. (**A**) Evaluation of the tumorigenicity of SW480 and SW620 cells. (Scale bar 100 μm). (**B**) Graph of the relative tumorigenicity of each cell line (*** *p* < 0.001).

**Figure 6 ijms-24-06476-f006:**
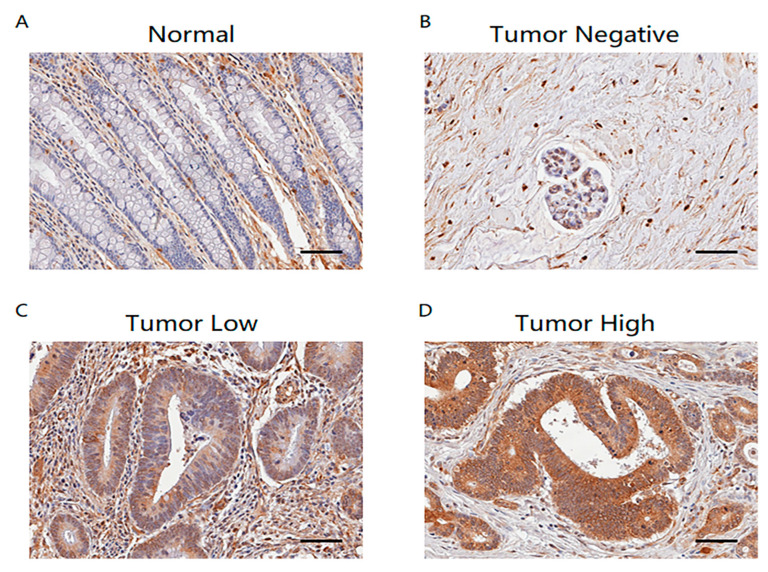
Confirmation of TSPAN5 expression in colon cancer tissue. (**A**) No expression in normal tissue. (**B**) No expression in colon cancer tissue. (**C**) Low expression in colon cancer tissue. (**D**) High expression in colon cancer tissue (Scale bar: 100 μm).

**Figure 7 ijms-24-06476-f007:**
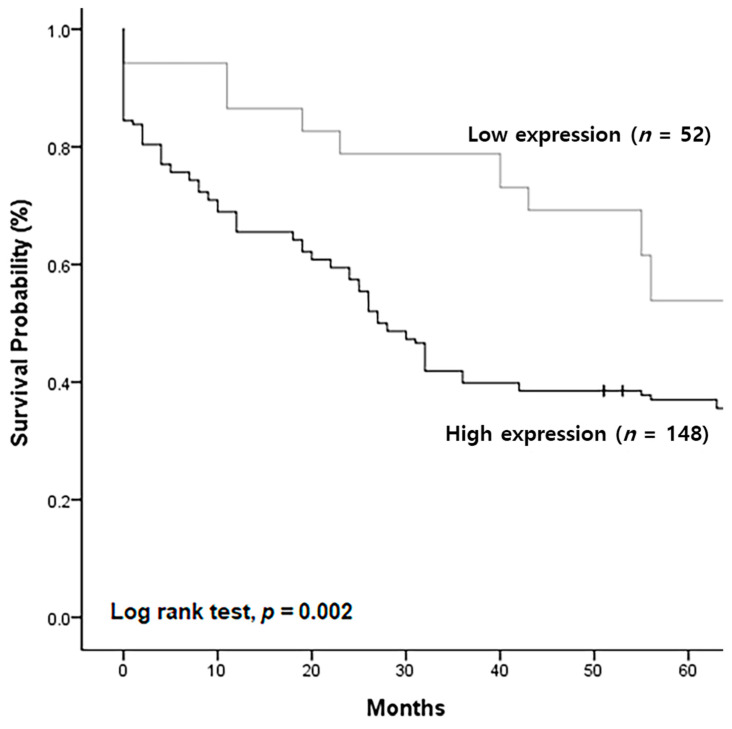
Changes in patient survival rate according to TSPAN5 expression level expressed by the Kaplan–Meier graph.

**Table 1 ijms-24-06476-t001:** Relationship between chi-square values for TSPAN5 expression and the clinicopathological factors of colorectal cancer.

Clinicopathological Factors	TSPAN5 Expression		Total (*n* = 200)	*p* Value
	Low (*n* = 52)	High (*n* = 148)		
Age, years, mean (SD)				
Sex, *N* (%)				0.034
F	10 (16.7)	50 (83.7)	60	
M	42 (30.0)	98 (70.0)	140	
pT, *N* (%)				0.085
1,2	5 (50.0)	5 (50.0)	10	
3,4	47 (24.7)	143 (75.3)	190	
pN, *N* (%)				<0.001
0	31 (43.1)	41 (55.9)	72	
1,2	21 (16.4)	107 (83.6)	128	
Metastasis, *N* (%)				0.077
Negative	38 (29.7)	90 (70.3)	128	
Positive	14 (18.4)	58 (80.6)	72	
Stage, *N* (%)				0.005
I and II	25 (38.5)	40 (61.5)	65	
III and IV	27 (20.0)	108 (80.0)	135	

**Table 2 ijms-24-06476-t002:** Univariate and multivariate Cox regression models.

Clinicopathological Factors	Variable	Univariate Analysis		Multivariate Analysis	
		Hazard Ratio (95% CI)	*p* Value	Hazard Ratio (95% CI)	*p* Value
Age	<0 years vs. ≥60 years	0.884 (1.644–1.214)	0.445	1.084 (0.760–1.546)	0.655
Sex	Female vs. Male	0.870 (0.620–1.220)	0.419	0.698 (0.479–1.017)	0.061
pT stage	T1, T2 vs. T3, T4	3.051 (1.249–7.455)	0.014	1.828 (0.693–4.822)	0.223
pN stage	N0 vs. N1, N2	2.051 (1.459–2.885)	<0.001	0.596 (0.320–1.109)	0.103
Metastasis	Negative vs. Positive	1.623 (1.159–2.271)	0.005	1.911 (1.608–2.363)	0.049
Clinical stage	I, II vs. III, IV	2.697 (1.864–3.903)	<0.001	4.078 (1.969–8.445)	<0.001
TSPAN5 expression	Low vs. High	1.763 (1.220–2.549)	0.003	1.528 (1.027–2.274)	0.036

## Data Availability

All data necessary to support the reported results are present in the main text of the article.

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
