# Peer review of "High Expression of Tetraspanin 5 as a Prognostic Marker of Colorectal Cancer"

_ijms, 2023, doi:10.3390/ijms24076476_

Round 1

Reviewer 1 Report

This research paper by Sanghyun Roh et al. titled " High expression of tetraspanin 5 as a prognostic marker of colorectal cancer" is an informative study that aims to examine the role of TSPAN5 in colorectal cancer recurrence and metastasis. The manuscript is well-structured and the objectives of the article are clearly stated in view of the subject. However, several  issues in this study need to be addressed:

Major issues:

·         Materials and Methods:

1.      It is recommended that the Authors report the incubation time for siRNA transfection assay. According to the RNAiMAX (Thermo Fisher, MA, USA) manufacturer the incubation time of the cells is 24-48h. Additionally, the Authors should explain if they used a scramble sequence for normalization. Generally, the Author could describe the siRNA transfection assay in more detail.

2.      It is strongly suggested that the Authors address the cDNA synthesis priming.

3.      In the part that the Authors describe the Western blot assay, they refer that “ A 10% gel was used for electrophoresis”. The Authors could clarify if they used agarose- or acrylamide- based gel.

·         Results:

4.      In bar plots of Figures 1, 2, 3 and 4, a label in the y-axis is missing. The Authors should fix this issue in order that the Figures be clearly presented and be more comprehensible. 

5.      The Authors could explain why they use PCR and not qPCR assay to confirm the relative expression levels of TSPAN5 in four colorectal cancer cell lines.

6.      It is advised that the Authors mention the non-transfected control groups. They used as control groups cells that were incubated only with lipofectamine or untransfected cells?

·         Discussion:

7.      Additional commentary could be added to the Discussion, regarding the importance of the findings and their potential utilization in standard clinical practice. The Authors are advised to be more critical in the content they include and be more focused on the most important parts of their study.

8.      The Authors should include relevant literature in this section, present the limitations of their study and how the limitations could be overcome.

·         The literature could be more up to date; more relevant and recent literature needs to be added.

Minor issues:

·         Materials and Methods:

9.      The siRNA sequence TSPAN5 (1) has an error. Please fix this issue.

10.  A table with all the primer sequences should be included.

11.  Proofreading of the manuscript is needed since there are several syntax and grammar errors throughout the text.

Author Response

Major issues:

  • Materials and Methods:
  1. It is recommended that the Authors report the incubation time for siRNA transfection assay. According to the RNAiMAX (Thermo Fisher, MA, USA) manufacturer the incubation time of the cells is 24-48h. Additionally, the Authors should explain if they used a scramble sequence for normalization. Generally, the Author could describe the siRNA transfection assay in more detail.

1A. We added some informations about siRNA treatment process.

  1. It is strongly suggested that the Authors address the cDNA synthesis priming.

2A.  We write about cDNA synthesis process for more detail.

  1. In the part that the Authors describe the Western blot assay, they refer that “A 10% gel was used for electrophoresis”. The Authors could clarify if they used agarose- or acrylamide- based gel.

3A.  We describe in more detail, “sodium dodecyl sulfate polyacrylamide gel” in material and method part.

  • Results:
  1. In bar plots of Figures 1, 2, 3 and 4, a label in the y-axis is missing. The Authors should fix this issue in order that the Figures be clearly presented and be more comprehensible. 

4A. We added the y-axis about the figure.

  1. The Authors could explain why they use PCR and not qPCR assay to confirm the relative expression levels of TSPAN5 in four colorectal cancer cell lines.

5A. This is a data comparing the expression amount of end point through PCR and electrophoresis. we use the conventional PCR, not qPCR because it was not comparing real-time amplification for the TSPAN5 gene.

  1. It is advised that the Authors mention the non-transfected control groups. They used as control groups cells that were incubated only with lipofectamine or untransfected cells?

6A. We did not use scramble siRNA, but we treated lipofectamine RNAiMAX in control cell to confirm the cytotoxicity of reagent.

  • Discussion:
  1. Additional commentary could be added to the Discussion, regarding the importance of the findings and their potential utilization in standard clinical practice. The Authors are advised to be more critical in the content they include and be more focused on the most important parts of their study.

      7A. Thank you for your advice. We added some comments in manuscript.

  1. The Authors should include relevant literature in this section, present the limitations of their study and how the limitations could be overcome.
  • The literature could be more up to date; more relevant and recent literature needs to be added.

      8A. We include some relevant literature in discussion part.

Minor issues:

  • Materials and Methods:
  1. The siRNA sequence TSPAN5 (1) has an error. Please fix this issue.

9A. We updated the sequence about TSPAN5 siRNA.

  1. A table with all the primer sequences should be included.

10A. All primers we used are described in “Material and Method 4.3. Reverse-transcription polymerase chain reaction”

  1. Proofreading of the manuscript is needed since there are several syntax and grammar errors throughout the text.

           11A. we appreciate your comment. We checked for the manuscript and change the errors.

Reviewer 2 Report

The manuscript Sanghyun Rohof et a. reports the possible use of TSPAN5 gene as a biomarker for CRC progression. In particular, the evaluation of TSPAN5 gene suppression, confirmed by statistical analyses, shows that it is related to CRC proliferation, migration, invasion and tumorigenesis. In light of this, evaluation of TSPAN5 expression may be helpful and may facilitate steps such as determination of an appropriate treatment method in patients with CRC.

In my opinion, this manuscript presents a suitable amount of new information to allow publication on IJMS. Nonetheless, some points need to be clarified.

1.     The raw image of Western blotting should be inserted in Figure 1C. The scientific journals don't accept collages anymore;

2.     Same request of point 1 for Figure 2C;

3.     In Figure 4A and C, the scale bar is not reported;

4.     In Figure 5A the scale bar is not reported;

5.     The authors in the discussion explicitly give information on the real number of colorectal cancer patient tissues examined. In my opinion, 200 colorectal cancer patient tissues are few to indicate the expression of the gene in question as a biomarker;

6.     In materials and methods, no specific paragraph giving information on the number of colorectal cancer patient tissue analyses is reported.

Author Response

  1. The raw image of Western blotting should be inserted in Figure 1C. The scientific journals don't accept collages anymore;

1A : We can suggest the raw data of the western blotting. We crop the band part for the figure, but it is not a collaged file.

2.     Same request of point 1 for Figure 2C; 

  1. In Figure 4A and C, the scale bar is not reported;

3A : Thank you for your advice. We make the scale bar in red color that you can see it well.

  1. In Figure 5A the scale bar is not reported;

4A : Thank you for your advice. We make the scale bar in red color that you can see it well.

  1. The authors in the discussion explicitly give information on the real number of colorectal cancer patient tissues examined. In my opinion, 200 colorectal cancer patient tissues are few to indicate the expression of the gene in question as a biomarker;

 5A : we use 5 of TMA slide, total 200 samples of colorectal patient.

  1. In materials and methods, no specific paragraph giving information on the number of colorectal cancer patient tissue analyses is reported.

6A : In immunohistochemistry section, we write about the information of TMA slide. We used these slides for IHC, and quantification was performed for statistical analysis.

Reviewer 3 Report

This is an interesting paper with a high potential to be recognized and cited by readers. Despite undoubted strengths, it needs improvement before final acceptance. The results are presented in detail and transparently, but the discussion is too short and simplistic. In the discussion, please consistently address all the results that you presented in the previous section. Please also discuss your results in comparison to other prognostic markers in colorectal cancer. One relevant publication was not included in the discussion (Qi et al. 2020, doi: 10.1186/s12935-020-01353-1).

Author Response

This is an interesting paper with a high potential to be recognized and cited by readers. Despite undoubted strengths, it needs improvement before final acceptance. The results are presented in detail and transparently, but the discussion is too short and simplistic. In the discussion, please consistently address all the results that you presented in the previous section. Please also discuss your results in comparison to other prognostic markers in colorectal cancer. One relevant publication was not included in the discussion (Qi et al. 2020, doi: 10.1186/s12935-020-01353-1).

  1. Thank you for your advice. We read the article and added some comments about the topics.

Round 2

Reviewer 1 Report

The manuscript has been significantly improved. All major issues have been addressed.

Reviewer 2 Report

In my opinion the manuscript has been improved and can now be accepted for publication if the editor agrees